# Investigating the Neural Bases of Risky Decision Making Using Multi-Voxel Pattern Analysis

**DOI:** 10.3390/brainsci12111488

**Published:** 2022-11-02

**Authors:** Yanqing Wang, Xuerui Peng, Xueping Hu

**Affiliations:** 1Institute of Psychology, Chinese Academy of Sciences, Beijing 100101, China; 2Department of Psychology, University of Chinese Academy of Sciences, Beijing 100049, China; 3Faculty of Psychology, Technische Universität Dresden, 01069 Dresden, Germany; 4School of Linguistic Science and Art, Jiangsu Normal University, Xuzhou 221009, China; 5Key Laboratory of Language and Cognitive Neuroscience of Jiangsu Province, Collaborative Innovation Center for Language Ability, Xuzhou 221009, China

**Keywords:** risky decision making, multi-voxel pattern analysis, risk preference, cognitive control

## Abstract

Choices between smaller certain reward and larger riskier reward are referred to as risky decision making. Numerous functional magnetic resonance imaging (fMRI) studies have investigated the neural substrates of risky decision making via conventional univariate analytical approaches, revealing dissociable activation of decisions involving certain rewards and risky rewards. However, it is still unclear how the patterns of brain activity predict the choice that the individual will make. With the help of multi-voxel pattern analyses, which is more sensitive for evaluating information encoded in spatially distributed patterns, we showed that fMRI activity patterns represent viable signatures of certain and risky choice and individual differences. Notably, the regions involved in representation of value and risk and cognitive control play prominent roles in differentiating certain and risky choices as well as individuals with distinct risk preference. These results deepen our understanding of the neural correlates of risky decision making as well as emphasize the important roles of regions involved in representation of value and risk cognitive control in predicting risky decision making and individual differences.

## 1. Introduction

Decision making under risk occurs on a regular basis in our daily lives. Although certain degree of risk-taking behavior is beneficial and necessary for the survival and evolution of humanity, excessive risk-taking may cause serious social and health issues, such as pathological gambling [1] and substance abuse [2]. Risk-taking behavior was usually investigated via two-alternative choice tasks, where participants were typically required to choose between a probabilistic reward option and a sure reward option [3,4,5,6]. For example, choose between the option that has a 60% chance of winning 40 dollars and the option that is certain to receive 20 dollars.

For decades, researchers have studied the neurobiological mechanisms supporting risk-taking behavior from the perspectives of neuroanatomy and neuron activation pattern [7,8,9]. With functional magnetic resonance imaging (fMRI), a distributed subcortical-cortical network mainly consisting of the striatum, amygdala, anterior cingulate cortex (ACC), dorsolateral prefrontal cortex (DLPFC), and ventromedial prefrontal cortex (vmPFC) was found to be involved in risky decision making [9,10,11,12,13,14,15]. Nevertheless, even though most existing works investigated the neural substrate of risk-taking behavior via conventional univariate analytical approaches, which look at the average level of activity in brain regions, resent researches suggested that specific decision making can be encoded in a spatially distributed way, which is reflected in voxel-level BOLD activation patterns. And elevated activity in a particular brain region does not necessarily indicate that relevant information is more effectively encoded or represented there. Meanwhile, this kind of distributed information can be measured with the help of multi-variate approach, which overcomes the limitations of univariate approach by searching for the optimal combination of scattered voxels and evaluating their contributions to decision discriminability [16,17,18].

Linear Support Vector Machine (SVM), a multi-variate algorithm, is often applied in the field of neural signal-based prediction [16,19,20,21]. Such a machine learning algorithm learns to classify observations into one of two categories from the training data. It is well-suited to detect subtle and spatially distributed differences over the entire brain among distinct cognitive states of decision making [22,23]. Despite the utility and advantage of such a method has been ascertained in many domains of systems in neuroscience, corresponding insights into risky decision making still remain scarce.

In this study, we combined fMRI with multivariate analysis techniques in a relatively large sample to measure spatial ensemble coding of different certain and risky choices. Moreover, risk preference varies greatly among individuals, on which different preferences are often exhibited even under similar scenarios. And individual differences in risk preference are associated with health behaviors [24] and susceptibility to mental illness [25]. Previous studies have identified the psychological characteristics associated with individual differences in risk preference [26,27]. Here, we also conducted analysis to shed light on whether the neural pattern could predict the difference between high and low risk preference participants.

## 2. Materials and Methods

### 2.1. Participants

Data used in this study were obtained from the OpenNeuro database with accession number of ds002843. The data set contains behavioral and brain imaging data from 166 healthy adults. More information about participant and study procedures can be found in the corresponding data paper [28]. All participants completed two fMRI testing sessions on two separate days, before and after cognitive training, respectively. At the pretraining session, participants were instructed to complete four runs of risky decision-making task. After the cognitive training, participants were instructed to complete another four runs at the second session to assess the effect of cognitive training. For the purpose of study, the current analysis just included data from the pretraining session.

After removing participants with missing files, extreme choice behavior (risk tolerance, α < 0.34 or α > 1.32) and large head motions (>3 mm), fMRI data from a total of 133 healthy participants (50 females, mean age SD = 24.35 ± 4.5, risk tolerance α = 0.68 ± 0.20) were used in the final analysis. All participants gave written informed consent following procedures approved by the University of Pennsylvania Institutional Review Board.

### 2.2. Risk Task

The detailed description of the risk task has been provided elsewhere. In brief, participants had to choose between a fixed small certain reward (100% chance of 20 dollars) and a larger riskier reward (e.g., 20% chance of 60 dollars), whose magnitude and probability varied from trial to trial. After a choice was made, choice feedback (1 s) was given to the participants. Participants completed a total of 120 choices inside the scanner.

### 2.3. Data Acquisition and Preprocessing

Functional MRI scanning were acquired with a 3.0-T Siemens scanner. Functional scans recorded using a standard echo-planar imaging (EPI) sequence of 53 axial slices (Repetition Time [TR] = 3000 ms, Echo Time [TE] = 25 ms, resolution matrix = 64 × 64, voxel size = 3 × 3 × 3 mm). T1-weighted scans were acquired using a standard Magnetization Prepared Rapid Acquisition Gradient Echo (MPRAGE) sequence (T1: 1100 ms; axial slices: 160 axial slices; matrix: 192 × 256). Additionally, a B0 field map was acquired (TR: 1270 ms; TE1: 5.0 ms; TE2: 7.46 ms) to support the off-line estimation of geometric distortion in the functional data.

Imaging preprocessing and statistical analyses were performed using SPM12 (https://www.fil.ion.ucl.ac.uk/spm12/; accessed on 1 March 2022). Functional images were slice-timing corrected and then realigned for motion corrections and unwarped using the gradient echo field maps. Realigned functional images were spatially normalized to the MNI template, and then smoothed with a 6mm full-width half-maximum Gaussian kernel.

### 2.4. Behavioral Data Analysis

Risk preference (α) in this experiment were estimated by fitting the following function:SV = p × Aα
where SV is the subjective value and p is the probability of winning amount A. For the risky option, there is always a 1 − p chance of winning nothing. Higher α indicates a larger risk preference and lesser degree of risk aversion. The risk preferences in the top 27% of the distribution were assigned to the high-risk preference group (*n* = 36, risk preference, α = 0.94 ± 0.11) whereas those in the bottom 27% to the low-risk preference group (*n* = 36, risk preference, α = 0.45 ± 0.05).

### 2.5. Univariate Activation

A general-linear model (GLM) was used to estimate task effects for each participant. The GLM had the following two regressors of interest: (1) trials in which the certain choice was chosen (certain choice); (2) trials in which the risky choice was chosen (risky choice). Task regressors were convolved using the canonical hemodynamic response function. The 6 head motion parameters (3 translation and 3 rotation parameters) were also included in the model as regressors of no interest. Data were high-pass filtered with a cutoff period of 128 s to remove low frequency signal drift. We computed first-level contrasts for the following: (1) certain choice, (2) risky choice, (3) risky choice–certain choice. The contrast maps were used in the following multi-voxel pattern analyses.

A leave one-subject-out cross validation procedure (i.e., exclude one participant for testing, train with the remaining) was used to estimate the predictive capability of neural encoding patterns on risky decision making.

### 2.6. Multi-Voxel Pattern Analysis

We conducted two classifiers which used the linear support vector machines (SVM, https://www.csie.ntu.edu.tw/cjlin/libsvm/; model setting: “-s 0 -t 0”; accessed on 1 March 2022) with default parameter (c = 1 [29]). Using the data of 133 participants, the first classifier was employed to assess the difference in the neural representation between certain choice and risky choice, which draw upon the contrast images of certain choice and risky choice extracted respectively for each participant, resulting in two images per participant. Using the data of low and high-risk preference group participants (*n* = 72), the second classifier was used to evaluate the difference of the neural representation between high and low risk preference participants in the contrast between certain choice and risky choice, which utilized contrast image of certain choice vs risky choice, resulting in one image per participant. For each classifier, a leave one-subject-out cross validation procedure (e.g., for the first classifier: exclude the data from one participant for testing, train with the remaining data from 133 participants) was performed to estimate the predictive capability of neural encoding patterns on risky decision making. To facilitate classification across participants, each feature (i.e., parametric value of each voxel) was normalized across the training set, and the normalization parameters were applied for normalizing the test set. The classifier was trained in the training set and then applied to the test set to obtain the category labels of untested participant. After completing all rounds of cross-validation, the average classification accuracy was calculated to quantify the performance of the classifier. To determine whether the accuracy was significantly higher than random level (i.e., 50%), we performed a permutation test by randomly shuffling the task labels 1000 times and running the above prediction pipeline for each time. Based on a null distribution of the accuracy, we estimated the significance by dividing the number of permutations that showed a higher value than the actual accuracy by the total number of permutations.

In addition, in order to identify the brain regions made most prominent contribution in the classifier, we trained a model using images from all participants (first classifier: 133 participants, second classifier: 72 participants) to obtain the regression coefficient (i.e., weight) for each voxel [30,31,32]. A higher absolute value of the regression coefficient indicates a greater contribution of the corresponding feature to the classification.

## 3. Results

### 3.1. Neural Pattern Differentiates Decision of Certain and Risky Choices

Using the data of 133 participants, the first classifier was employed to assess the difference in the neural representation between certain choice and risky choice. The results showed that the classifier was able to efficiently differentiate individuals’ risky decisions between certain reward options and the risky ones with classification accuracy as high as 70.30% (*p* < 0.001, 1000 permutation tests). The top 1% high weighted voxels located in the bilateral striatum (MNI coordinate of peak voxel: −10, 8, −4; 12, 8, −6), ventromedial prefrontal cortex (vmPFC, −2, 38, −10; −34, 42, −10), anterior cingulate cortex (ACC, 0, 42, 8), mid-cingulate cortex (MCC, 0, −32, 36), bilateral superior parietal lobe (SPL, −44, −46, 46; 44, −40, 50), and left dorsolateral prefrontal cortex (dlPFC, −42, 38, 8) (Figure 1), indicating these regions play crucial roles in the representation of one’s risky choices.

### 3.2. Neural Patterns Distinguish Participants with High and Low Risk Preference

Substantial individual differences of one’s risk preference have been widely reported. Here, the 133 participants were ranked in order of their risk preference (α), the risk preference in the top 27% of the distribution were assigned to the high-risk preference group (*n* = 36) whereas those in the bottom 27% to the low-risk preference group (*n* = 36). We then performed second classification to discriminate the same contrast images (certain choice vs. risky choice) from high and low risk preference individuals. The results showed that albeit the decision contrasts were held invariant, the classifier can robustly classify the contrast image into the relevant groups (ACC = 94.44%, *p* < 0.001, 1000 permutations). The top 1% high weighted voxels situated in the bilateral insula (−30, 22, −4; 38, 22, −4), bilateral SPL (−26, −56, 52; 32, −56, 46), bilateral dlPFC (−42, 4, 28; 48, 8, 28) and ACC (0, 20, 44) (Figure 2). Specifically, classifications conducted with these regions as ROIs revealed that each region alone can categorize high and low impulsive participants successfully (*p*s < 0.015) with high accuracies (mean accuracies > 68.06, Table 1).

## 4. Discussion

Using multi-voxel pattern analysis, the present study found that the spatially distributed information of neural activity in human brain can not only robustly predict one’s decision choice in risky decision making, but also successfully differentiate high and low risk preference participants with high accuracies. These findings provide strong evidence to confirm the predictive role of the general neural encoding patterns for individual risky decisions in the content of the brain-behavior domain.

Although previous researches have extensively studied the dissociable neural mechanisms between different choices processes in risky decision making [27,33,34,35,36,37], very few studies have examined such different neural mechanisms from the perspective of neural representation. By using the relatively raw data and taking full advantage of information from the voxels, multi-voxel pattern analysis endows us with higher sensitivity and reliability [38]. Our study benefited from such advantages, as besides confirming dissociable patterns in regions previously reported to have different activation levels, we identified a range of regions with distinct distributed ensemble activities that have been documented as overall equally activated.

The classifier showed that neural representation could distinguish the certain and risky choices with high classification accuracy at the whole brain level. This result suggests that despite that the physical inputs were the same, different choices were accompanied with decodable neural representation patterns. Specially, by quantifying feature weight for each voxel, we found that several regions of cortico-striatal network with high weights play a vital role in the classification, such as the vmPFC, striatum, SPL, ACC, MCC and dlPFC. The high weight region in the classification reflects the effectiveness of encoding and representation of decision choices information in this region. Noticeably, the value and risk of an option along with the agent’s risk aversion are the basic factors implicated in the risky decision behavior [9]. These regions have been implicated in such factors processes. For example, activity in vmPFC and the striatum reflects an integration of the magnitude and probability of rewards for given risky options [9,11,36,39,40,41]; activity in the SPL reflects the probability of outcomes [34,35]; activity in the ACC is associated with the volatility of reward environment [12] and the variability of expected outcomes [13,37,42]; and activity in the MCC and dlPFC reflect the engagement of cognitive control to suppress the risk raking behaviors [14,27,43]. Our results corroborated these findings and took a step further with the help of multi-voxel pattern analysis to demonstrate that the adequate information encoded in these different brain regions can largely determine what decision would be made in the choices.

Considerable individual differences have been repeatedly described in risky decision making [6,8,37]. Brain structure and function studies have investigated the neural correlates of risky decision making in different risk preference individuals and showed that they exhibited various neural dynamics even under identical choice conditions. For instance, increased gray matter volume in the right posterior parietal cortex is related to increased risk preference [4]. In line with these works, our results indicate that although decision contrast remained the same (risky choice vs. certain choice), drastically distinct patterns were found in individuals with different risk preference, indexed by high accuracy when decoding high and low risk preference participants. Furthermore, we found that the neural pattern of ACC, DLPFC, SPL and insula provides sufficient discriminative information to distinguish high and low risk preference participants. Besides the functions of ACC, DLPFC, SPL and insula in risky processing mentioned above, such regions are also important nodes of cognitive control network [44,45]. Moreover, high risk preference reflects individual impulsivity [46], and the research on the mechanism of decision impulsivity emphasizes the engagement of cognitive control to reduce choice impulsivity [47,48]. Thus, we suggest that besides the representation of value and risk of option information during risky decision making, cognitive control takes vital roles in differentiating the representations of high and low risk preference participants.

The current results have a number of limitations that should be noted. First, the risk-reward probability of each trial is certain, but the risk taking in real life involves multiple components, such as unknown risks, loss aversion and learning of implicit probability structures [3]. Future research will need to clarify the relationship between changes in neural activity patterns and such components of risk-taking behavior. Second, this study only includes younger and healthy adults. Previous research found that older adults showed higher risk aversion compared with younger adults [49]. Moreover, the age-related changes in risk preference were related to changes in right posterior parietal cortex grey matter volume [4]. Future research will need to clarify the relationship between changes in neural activity patterns and the changes in age-related risk preference. Meanwhile, a large body of research links maladaptive risk preference and psychiatric illnesses such as substance abuse and gambling disorder, making it being proposed as a candidate behavioral marker for psychopathology [50]. Future research will need to clarify the relationship between changes in neural activity patterns and maladaptive risk preference of psychiatric illnesses. In addition, our study identified functional neural signatures in cognitive control network corresponding to one’s risk preference, which underscores cognitive control as potential mechanism underlying various maladaptive risk preference related disorder, making these regions promising therapeutic target for altering high risk preference behaviors. Previous study draws upon repetitive transcranial magnetic stimulation and found that disruption of function of the lateral prefrontal cortex displayed significantly riskier decision making [15]. Future research could adopt cognitive control training as a candidate therapy to ameliorate the maladaptive risk preference in the clinical population.

## 5. Conclusions

In summary, this study adopted multi-voxel pattern analysis approach in a relatively large sample data to explore the underlying neural substrates of risky decision making. At the global level, the classifier built with global activation pattern not only successfully distinguished the certain and risky choices, but also effectively distinguished the high- and low-risk preference participants. Furthermore, the results underline the regions in representation of value and risk of options and cognitive control as critical components of these neural signatures. These results deepen our understanding of the neural correlates of risky decision making as well as emphasize the utility of pattern analysis in predicting risky decision making and individual differences.

## Figures and Tables

**Figure 1 brainsci-12-01488-f001:**
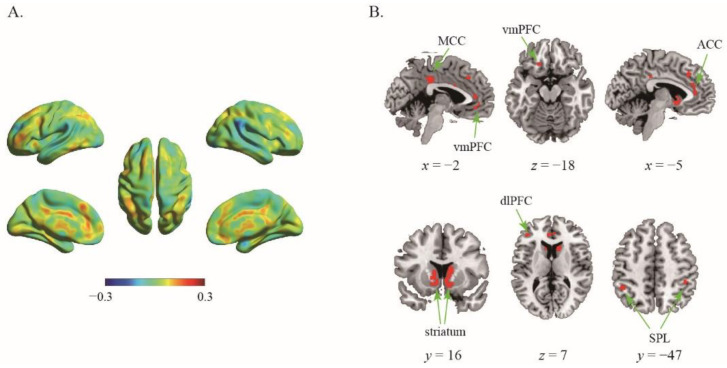
The results of first classifier. (**A**) The whole-brain weighted map. The color bar indicates weight value; (**B**) The top 1% voxels with highest weights (absolute value).

**Figure 2 brainsci-12-01488-f002:**
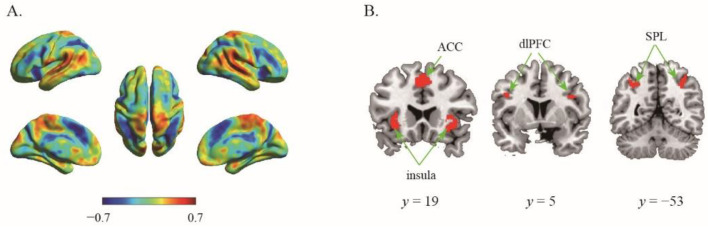
The results of second classifier. (**A**) The whole-brain weighted map. The color bar indicates weight value; (**B**) The top 1% voxel with highest weights (absolute value).

**Table 1 brainsci-12-01488-t001:** Decoding accuracies of high and low risk preference participants in top 1% high weighted areas.

Region	Cluster Size	Accuracy	*p*
ACC	525	76.39	<0.001
Left DLPFC	35	73.61	<0.001
Left insula	172	79.17	<0.001
Left SPL	272	68.06	=0.008
Right DLPFC	83	68.06	=0.015
Right insula	209	75.00	<0.001
Right SPL	175	55.56	=0.18

## Data Availability

Data supporting the findings of this study are available on OpenNeuro database (accession number: ds002843).

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
