# Peer review of "Investigating the Neural Bases of Risky Decision Making Using Multi-Voxel Pattern Analysis"

_brainsci, 2022, doi:10.3390/brainsci12111488_

Round 1
Reviewer 1 Report
The paper aims to further identify, by multi-voxel pattern analysis, the neural basis of risky decision making. The authors use data from an existing cohort of healthy subjects and apply their analysis in fMRI. The overall approach is original and relevant, especially with respect to current research issues on this subject.
Some points deserve to be clarified in the manuscript to increase the potential of the article.
First of all, the decision making task chosen is under risk, and thus, obviously, without learning effect. It corresponds to a decision where the consequences are known, which is a special case of decision making. In real life situations, risk-taking behaviors correspond rather to under-ambiguity decision making (unknown consequences of the choice), which solicits different neural processes. It seems that the "behavioral" section could go into more detail on this aspect.
In the methodological section, again, it would be relevant to add a calculation of the number of participants needed. Indeed, since the authors not only identify the activations, but also weight their impact in the decision-making processes, it seems necessary to detail on which basis the 133 participant data are sufficient. It is, in fact, mentioned the use of 133 participants as 1st classifier and 79 as 2nd classifier. Unless I am mistaken, these elements are not justified.
The result section is clear and explicit.
The discussion section lacks critical analysis. Indeed, the results are very well synthesized, reinforced by the known literature, and the potential of this research is enhanced. Nevertheless, for specialists in the field, it quickly makes sense. It therefore seems rather necessary to reinforce the limits of the work, which are as many avenues of further research, and which are integrated in a more global perspective of progress of the work on the field.
Reviewer 2 Report
This is a well written manuscript with novel content. Although it may not have high clinical impact, these findings are surely worth publishing. Research methodology is rigorous and the sample size is sufficiently large, which is indeed a plus for neuroimaging studies.
The authors should just correct minor grammatical/punctuation errors and include more information (reference more relevant studies) about the relationship between risk taking behaviour and addictions (behavioural and pharmacological) in the introduction and discussion sections.
Reviewer 3 Report
The authors present an interesting topic about medical imaging analysis using voxel analysis. The topic is of international interest and the analysis elements at the voxel level were carried out in different research laboratories.
Also fMRI equipment can determine mental activity using voxel analysis.
The authors present in the article a voxel-level analysis of fMRI images, but use the basic methods - algorithms - presented in the specialized literature for this type of analysis (fMRI images, SVM, cluster). A first question: what is the novelty contribution of the presented method?
Francisco Pereira, a researcher at Princeton University, said in an article "fMRI is a brilliant tool for generating these correlations, allowing you to see the associated activity in groups of neighboring voxels - which correspond well to brain regions - as well as activity in more distant parts of the brain".
To improve the article, the authors must clarify the following:
1. what differentiates their method in relation to the relevant articles from the specialized literature that deal with the same subject?
2. are the results obtained by the authors at least equal / superior to the results reported in the specialized literature (see the quote above)?
3. what computing power is needed to perform the multi-voxel analysis?
4. what are the concrete contributions of the authors for the proposed topic - voxel analysis?
I ask the authors to clarify the answers to the mentioned suggestions for improving the article.
Round 2
Reviewer 1 Report
The revision made by the authors is satisfactory and has improved the manuscript.
Reviewer 3 Report
The authors have improved the article, it can be published in this form.